# The Influence of Pharmacogenetics on the Clinical Relevance of Pharmacokinetic Drug–Drug Interactions: Drug–Gene, Drug–Gene–Gene and Drug–Drug–Gene Interactions

**DOI:** 10.3390/ph14050487

**Published:** 2021-05-20

**Authors:** Martina Hahn, Sibylle C. Roll

**Affiliations:** 1Klinik für Psychiatrie, Psychosomatik und Psychotherapie des Universitätsklinikums Frankfurt, 60528 Frankfurt, Germany; 2Dr. Amelung Privatklinik, 61462 Königstein, Germany; 3Klinik für Psychische Gesundheit, Klinikum Frankfurt Höchst, 65929 Frankfurt, Germany; Sibylle.Roll@KlinikumFrankfurt.de

**Keywords:** drug–drug interactions, drug–gene interactions, drug–g–gene interactions, phenoconversion, pharmacogenetics

## Abstract

Drug interactions are a well-known cause of adverse drug events, and drug interaction databases can help the clinician to recognize and avoid such interactions and their adverse events. However, not every interaction leads to an adverse drug event. This is because the clinical relevance of drug–drug interactions also depends on the genetic profile of the patient. If inhibitors or inducers of drug metabolising enzymes (e.g., CYP and UGT) are added to the drug therapy, phenoconcversion can occur. This leads to a genetic phenotype that mismatches the observable phenotype. Drug–drug–gene and drug–gene–gene interactions influence the toxicity and/or ineffectivness of the drug therapy. To date, there have been limited published studies on the impact of genetic variations on drug–drug interactions. This review discusses the current evidence of drug–drug–gene interactions, as well as drug–gene–gene interactions. Phenoconversion is explained, the and methods to calculate the phenotypes are described. Clinical recommendations are given regarding the integratation of the PGx results in the assessment of the relevance of drug interactions in the future.

## 1. Intoduction

Drug–Drug-Interactions (DDI) are a well-known cause of adverse drug events [1,2,3]. The number of potential drug interactions increases exponentially with the number of taken drugs, making it hard to consider all drug interactions in polypharmacy patients. Drug interaction databases can help the clinician to recognize and avoid adverse drug interactions [4]. A phenomenon can be observed in daily practice, though: even if a database warns about a clinically relevant DDI, the patient may not show any signs of a given DDI. This discrepancy between scientific evidence and clinical reality causes alert fatigue and also raises conflicts between the DDI warning pharmacists and no adverse drug event-observing physicians [5].

Pgx testing has become more popular over the last decade, but it is not yet a routine test, except in oncology. However, there is growing evidence that large proportions of patients are affected by an actionable genotype—a genotypes where a change in prescribing may be indicated. The evidence for Drug–Gene-Interactions (DGI) exists for many drug–gene pairs already. In pediatric patients, for example, the annual prescribing prevalence of at least one level A drug (recommendations for drugs available with high evidence regarding a particular genotype) ranges from 7987 to 10,629 per 100,000 patients [6]. Turner analyzed the data from non-ST elevation myocardial infarction patients (*n* = 1456) and found that 98.7% of the patients had at least one actionable genotype [7]. Within the interaction cohort (drug use and actionable genotypes available), 882 interactions were identified in 503 patients (77.1%), of which 346 interactions in 252 patients (38.7%) were substantial: 59.2%, 11.6%, 26.3%, and 2.9% substantial interactions were DDIs, DGIs, Drug–Drug–Gene-Interactions (DDGIs) and Drug–Gene–Gene-Interactions (DGGIs), respectively (see Table 1 for definitions). This shows that preemptive PGx testing could lead to a reduction in the clinically relevant interactions, especially in patients with polypharmacy.

There are several organizations, such as the Dutch Pharmacogenetics Working Group (DPWG), the Clinical Pharmacogenetics Implementation Consortium (CPIC), the Canadian Pharmacogenomics Network for Drug Safety (CPNDS), and the French National Network (Réseau) of Pharmacogenetics (RNPGx), that have developed clinical guidelines to help the clinician translating the scientific findings into clinical treatment recommendations. As we understand more about DDGIs and DGGIs, the influence of genetic variants on the drug level is becoming clearer and better understood. A problem in prior studies was that clinical genotype-focused association studies did not take pharmacokinetic DDIs into consideration. In fact, such studies only correlated the genotype with clinical outcomes. Typically, those studies focused on the drug metabolizing enzyme (DME) genotypes of the study population and assumed that genotypes of all the study subjects predict their functional phenotype. Despite wide inter- and intra-genotype variability in the metabolic capacity, clinical responses are thought to be simple binary outcomes for genotype groups. The aim of these studies was to examine the strength of the associations between genotype and clinical phenotype and, if shown to be strong, to develop a dosing regimen appropriate to each genotype. By not taking DDIs, DDGIs, and DGGIs into consideration, they may have missed clinically strong pharmacogenetic associations, thus compromising any potential for advancing the prospects of personalized medicine. Many studies, however, were not able to find those strong associations, which lead to the argument that PGx is not valuable in clinical practice.

Currently, most guidelines on DDIs neither consider the potential effect of genetic polymorphisms in the strength of the interaction nor do they account for the complex interaction caused by the combination of DDI and DGI when there are multiple biotransformation pathways, referred to as DGGI. Not surprisingly those guidelines often contradict each other [8]. Therefore, all the studies with only PGx results, as well as studies in which only DDIs were reported, are probably incorrect and will, thus, lead to false conlusions and potentially damaging clinical recommendations. Therapeutic drug monitoring (TDM)—a gold standard to monitor pharmacokinetic DDIs—is very helpful but cannot avoid adverse drug reactions in the beginning of a drug therapy, because TDM can only be carried out after five half-lifes of the drug when a steady state is reached.

Taken together, this has led many clinicians to the conclusion that PGx testing is not reliable. There is an uncertainty about when and whom to test, and few guidelines actually give a clinical recommendation in relation to this. It is important to understand the route of the drug through the body (LADME principle) and then understand how a SNP influences the absorption, distribution, metabolism, and excretion of the drug. We need “the whole picture” of the patient with all the comedication to predict the efficacy and tolerability of a drug.

In this review, we provide a framework for the classification of pharmacokinetic DDGIs and DGGIs caused by different mechanisms, and their potential impact on drug serum levels in the context of polypharmacy.

## 2. Pharmacokinetic Drug–Drug Interactions

In the past, the magnitude and clinical importance of drug interactions was often neglected [9] because of overalerting DDIs in clinical practice [10,11,12]. The CYP-inhibitory or -inducing potential of a drug and the substrate specificity of the potential victim drug must be considered (Table 2). CYP perpetrator drugs are inhibitors or inducers of CYP enzymes. CYP perpetrator drugs affect the metabolism of victim drugs (CYP substrates) and lead to an increase or decrease in serum concentrations of the victim drugs. The Food and Drug Administration categorizes CYP inhibitors or inducers as strong, moderate, or weak based on pharmacokinetic DDI studies [13]. CYP inhibitors that were categorized as “strong” (> fivefold increase in area under the plasma concentration–time curve (AUC) or > 80% decrease in clearance) or “moderate” (> twofold but < fivefold increase in the AUC or 50–80% decrease in clearance) and CYP inducers categorized as “strong” (≥ 80% decrease in AUC) or “moderate” (50–80% decrease in AUC) can be regarded as clinically relevant. Several tables of drugs as substrates, inhibitors, and inducers of drug-metabolizing CYP enzymes have been published in the past [14,15]. The categorization into strong, moderate, and weak allows a first grading into “relavant” or irrelevant “pharmacokinetic DDIs. However, in clinical practice, this classification is not always correct, and it may be related to genetic polymorphisms and DGIs, DDGIs, and DGGIs, as described in the next sections.

## 3. Drug–Gene Interactions

A genetic polymorphism is a variation in DNA sequence that leads to a reduced or increased activity of a DME. There are several different types of polymorphisms that can affect DGIs:

Single-Nucleotide Polymorphisms (SNPs), with a change of a single base pair, insertions or deletions, a variable number of tandem repeats (VNTR), and copy number variants (CNV), where the number of copies varies beween individuals [16]. SNPs can cause:(a)a change in the codon, which might change the amino acid that is transcribed;(b)a premature stop codon (no functional protein is formed);(c)different intron and exon splice junctions (no functional protein is formed);(d)an alteration in the stability of the mRNA (no proteins are formed);(e)a change in enhancer activity (gain of function);(f)or even no discernible consequence.

It is important to emphasise that SNPs—the change of a single base—can lead to a “loss of function”, “a decrease in funtion”, “a gain of function”, or even to no formation of DME at all, to understand phenoconversion in the later sections. From the definition, one can distinguish a haplotype (one allele, e.g., *1) from a diplotype (two alleles, e.g., *1/*41). Now, a genotype is the DNA sequence of an organism at a specific, defined location. In PGx, the term is used to describe a certain gene, e.g., CYP2D6 (Figure 1). The genotype could be an IM/PM (intermediate metabolizer/poor metabolizer). A phenotype, in comparison, is the obervable trait (e.g., blue eyes) or enzyme activity score of the DME in a patient. The genetic phenotype of DME can be divided into a poor, normal, intermediate, rapid, and ultra-rapid metabolizer status, taking both genotypes into consideration, e.g., IM/PM genotypes equals the IM phenotype; NM/IM genotype equals the NM phenotype. This takes both inherited genotypes into account (Figure 1). The phenotype can be observed by using TDM.

Many studies have shown that the efficacy and risk of side effects of a drug treatment is influenced by genetic variants of the DMEs and the transporters. Evidence-based guidelines with pharmacotherapeutic recommendations for combinations of specific drugs and genotypes or predicted phenotypes are essential for implementing acquired pharmacogenetic knowledge in daily clinical practice. The DPWG CPIC, CPNDS, and RNPGx have developed guidelines according to the genetic phenotypes of the patient. They also published a study to standardize the genotype-to-phenotype translation [17]. The impact of 440 genetic variants on the pharmacokinetics and pharmacodynamics of drugs are available at the CPIC website: www.cpicpgx.org, accessed on 29 March 2021.

## 4. Genetic Polymorphisms of DME of Phase I Metabolism

Several genes encoding cytochrome P450 enzymes are highly polymorphic, especially CYP2A6, CYP2B6, CYP2C8, CYP2C9, CYP2C19, CYP2D6, and CYP3A5. This requires a translation of genotypes into predicted genetic phenotypes. For DMEs, we know four different activities levels of the enzymes: normal function, decreased function, loss of function, and increased function.

This results in five different genetic phenotypes: normal metabolizer (NM), intermediate metabolizer (IM), poor metabolizer (PM), rapid metabolizer (RM), and ultra-rapid metabolizer (UM).

Studies have shown that DGIs change the efficacy and adverse drug event rate. Over the last decades, especially in relation to drugs with a narrow therapeutic index or prognosis, changing drugs like warfarin (*CYP2C9* genotype and risk of hemorrhage or stroke), tamoxifen (*CYP2D6* genotype and risk of therapeutic failure), clopidogrel (*CYP2C19* genotype and risk of thrombotic cardiovascular outcomes), irinotecan (*UGT1A1* genotype and risk of myelosuppression), and thiopurines (*TPMT* genotype and/or phenotype and risk of myelosuppression) were a focus of research. FDA label changes were conducted, and preemptive genotyping was recommended before starting drugs with a high toxicity and, therefore, a high risk for the patient to suffer from a severe adverse drug reaction. DPWG gives specific clinical recommendations for 54 DGIs already (see http://upgx.eu/guidelines, accessed on 29 March 2021).

Of all the DMEs, CYP2D6 and CYP2C19 are the most polymorphic. If there is a decreased activity (intermediate and poor metabolizers), a higher drug concentration increases the risk for adverse drug events and toxicity. If there is an increased activity (rapid and ultra-rapid metabolizers), a lower drug concentration is reached, and this results in a risk for therapeutic failure. These enzymes play a prominent role in the drug metabolism of psychotropic drugs, e.g., antidepressants, antipsychotics, and nonstimulants like atomoxetine but, also, opioids, betablockers, and proton-pump inhibtors. Adverse events, response failures, and/or medication nonadherence are common in patients receiving medications for the treatment of mental illness. This might be caused by DGIs in these patients. In psychopharmacotherapy, the CPIC and DPWG have developed guidelines and give treatment recommendations for SSRIs, TZAs, atomoxetin, and opioids, among others, in regards to the patient phenotypes of CYP2C19 and/or CYP2D6.

For CYP enzymes, there is a genotype acitivity score definition (Table 3). Depending on the two scores of each allele, the phenotype can be calculated, e.g., *4/*1 equals an activity score of 0 + 1 = 1. The patient’s genetic phenotype would be intermediate metabolizer (Figure 1).

Pharmacokinetic Drug–Gene-Interactions influence the serum concentration of the drug. Chang et al., for example, showed that there was an increase in the escitalopram levels of 95% and 30% in the poor and intermediate metabolizers of CYP2C19 compared to normal metabolizers, respectively. Rapid and ultra-rapid metabolizers were demonstrated to have a decrease in the escitalopram levels of 13% and 36%, respectively [18]. This is an important finding, because only 13% of psychiatric inpatients with a major depressive disorder are normal metabolizers of both CYP2C19 and CYP2D6 [19]. Hicks et al. provided specific recommendations in the CPIC guidelines on the dosing and use of SSRIs and tricyclics for each phenotype [20].

Clopidogrel, an antiplatelet agent that reduces the risk of myocardial infarction (MI) and stroke in patients with acute coronary syndrome (ACS), and in patients with atherosclerotic vascular disease, is another common example for DGIs. Clopidogrel is also indicated in combination with aspirin in patients undergoing percutaneous coronary interventions (PCI), e.g., the placement of a stent. The effectiveness of clopidogrel depends on its conversion to an active metabolite by CYP2C19. Individuals who carry two nonfunctional copies of the CYP2C19 gene are classified as CYP2C19-poor metabolizers. They have no enzyme activity and cannot activate clopidogrel via the CYP2C19 pathway, which means the drug will have no effect. Approximately 2% of Caucasians, 4% of African Americans, and 14% of Chinese are CYP2C19-poor metabolizers. Given the strong evidence for this DGI, the 2017 FDA-approved drug label for clopidogrel includes a boxed warning concerning the diminished antiplatelet effect of clopidogrel in CYP2C19-poor metabolizers. The warning states that tests are available to identify patients who are CYP2C19-poor metabolizers and to consider the use of another platelet P2Y12 inhibitor in patients identified as CYP2C19-poor metabolizers. CYP2C19-poor and -intermediate metabolizers also have an increased risk for ischemic stroke, myocardial infarction, and coronary angioplasty [21].

DGIs also play an important role in oncology therapies/treatment. Adjuvant tamoxifen therapy reduces breast cancer mortality by 31%. However, the tamoxifen effectiveness varies widely between individuals. Tamoxifen is a drug that requires metabolic activation by CYP2D6 to elicit its full pharmacologic activity, because the corresponding metabolites 4-OH-tamoxifen and endoxifen exhibit much higher binding affinities to the estrogen receptor than the parent compound. A study by He et al. showed that both poor and ultra-rapid metabolizers have a worse prognosis and a higher mortality compared to normal metabolizers when given tamoxifen. Poor metabolizers have a decreased efficacy, because the 4-OH-tamoxifen and endoxifen are not formed. Ultra-rapid metabolizers have more adverse drug events and a higher risk for discontinuation of tamoxifen and, in consequence, a worse prognosis [22].

For phase 1 enzymes, more adverse drug reactions in CYP2D6 ultra-rapid metabolizers are also documented for codeine (respiratory failure). Thus, especially for prodrugs, a fast metabolism to the active metabolite can cause toxicity. Examples of the substrates, inhibitors, and known genotypes are listed in Table 2.

## 5. DMEs of Phase 2 Metabolism

Some drugs undergo a metabolism in phase 2. In addition, there are several genetic polymorphisms known for these DMEs. There are a few examples that exist, revealing that the metabolites are more toxic than the parent drug, e.g., estradiol-17 glucuronide causes more adverse drug events than estradiol itself. Usually, the inhibition of phase 2 DMEs results in an increase in the serum levels and, therefore, a higher toxicity. Examples of substrates, inhibitors and inducers can be found in Table 4.

## 6. Drug Transporters (Phase 3)

Even at “normal” serum levels, drugs can show no efficacy, because they are unable to cross certain barriers in the body (e.g., the blood–brain barrier), or they can be highly toxic if they accumulate intracellularly (e.g., tenofovir-induced tubular toxicity in patients with an *ABCC2* polymorphism). By only measuring the serum concentration (TDM), those changes in the pharmacokinetics of the drug cannot be identified (and not be avoided). Hence, PGx testing can help to avoid ADRs and increase the efficacy, ideally in combination with TDM.

There are two major transporter superfamilies: the ABC transporter and the SLC transporter, consisting of the ABC transporter and SLC transporter families (e.g., ABCC family or SLC21 family) (Table 5). They play an important role in the absorbtion (intestine), distribution (blood–brain barrier, placenta, and testes), metabolism (liver), and excretion (biliary and renal) processes. Genetic variants of the transporters modulate the absorption, distribution, metabolism, and excretion of the drugs. A prominent example is statin-induced myopathy in carriers of the OATP1B1*5 polymorphism. Furthermore, bile salt export pump polymorphisms are known to cause drug-induced liver injury (DILI), which accounts for 20–40% of all hepatic failure cases observed worldwide [23]. A polymorphism of the multidrug-resistant protein type 2 (MRP2) increases the kidney toxicity of tenofovir administration [24].

Additionally, in epileptic patients, a transporter was found to be responsible for drug therapy resistance. About 30% of epileptic patients do not respond to Pg-P substrates but do if a Pg-P inhibitor is added to drug therapy [25,26], an example of how relevant phenoconversion can be in clincal practice if employed on purpose. The same effect is proposed for antidepressants and mood stabilizers [27,28,29].

It has been estimated that half of the patients above 65 years will use at least one drug for which PGx guidelines are available during a four-year period, and 25–33% will use two or more of these drugs [30].

## 7. Drug–Gene–Gene Interactions (DGGIs)

DGGIs can explain why, even if we consider all known DGIs and DDIs, there are serum levels that we cannot explain by those two factors alone. If two or more CYP enzymes metabolize a drug, then inhibition of one of these enzymes alone (by drug or genotype) may have minimal effect, due to redundancy of the pathways (Figure 2).

If a patient is an intermediate metabolizer for CYP2D6 and CYP2C19, a dose reduction might be indicated, while it is not recommended in CYP2D6 NM and CYP2C19 IM. Thus, the genotype of the second enzyme also influences the overall metabolism and, in consequence, the serum level. For tricyclic antidepressants, the CPIC guideline gives specific recommendation in a cross-table for each CYP2D6 and CYP2C19 genotype combination [20]. The same guidelines exist for thiopurines with the *TMPT* and *NUDT15* genotypes [31] and statins with the *SLCO1B1* and *ABCG2* genotypes [32]. This underlines how important the testing of a panel rather than a single DME genotype is. Bousman et al. recommended a testing panel for psychiatry that covers most of the DMEs. DPWG recommends a PGx passport that encompasses 58 variant alleles within 14 pharmacogenes (CYP2B6, CYP2C9, CYP2C19, CYP2D6, CYP3A5, DPYD, F5, HLA-B, NUDT15, *SLCO1B1*, TMPT, UGT1A1, and *VKORC1*) and can be used to optimize the pharmacotherapy for 49 commonly prescribed drugs throughout a patient’s lifetime [33].

In drugs that utilize more than two pathways, predicting the drug serum level becomes even more complicated. This complexity shows the importance of PGx-specialized clinical pharmacists to help interpret the results, taking drug interactions, as well as lifestyle factors, into account when giving recommendations. CPIC guidelines recommend the consultation of a clinical pharmacist if actionable genotypes are discovered. Many universities now offer PGx training and specializations for pharmacists.

How do genetic polymorphisms and drug interactions influence each other?

## 8. Drug–Drug–Gene Interactions (DDGIs) and Phenoconversion

Phenoconversion is the conversion of a genetic phenotype (i.e., PM, IM, NM, and UM) into a different phenotype by comedication or other nongenetic factors and is quite common [32]. Phenoconversion is a complex phenomenon that leads to genotype–phenotype mismatching without any genetic abnormality. It is particularly well-characterized for cytochromes P450 2D6 and 2C19. Although transient, phenoconversion can have a significant impact on the analysis and interpretation of genotype-focused clinical outcome correlations and in forensic toxicology conclusions [34] but, also, in everyday clinical practice. Phenoconversion resulting from nongenetic extrinsic factors has a significant impact on the analysis and interpretation of genotype-focused clinical outcome association studies and, ultimately, to the personalization of therapy in routine clinical practice. Having the genotype data available can help identify those nongenetic factors, which may lead to a decreased risk for the patient to suffer from adverse drug reactions by following a different treatment algorithm (e.g., order TDM, treat the infection, avoid the drug interaction, etc). Examples of nongenetic factors include inflammation, cancer, age, liver disease, and renal dysfunction [35].

The high phenotypic variability or genotype–phenotype mismatch, frequently observed due to phenoconversion within the genotypic NM population, means that the real number of phenotypic PM subjects is likely to be greater than predicted from their genotype alone. This is because many genotypic NMs would be phenotypically PMs [36]. Mostafa et al. analyzed an Australian cohort and found an increase in actionable genotypes due to phenoconversion in a large proportion of patients. The number of CYP2D6 PM increased from 5.4% (genotype predicted) to 24.7% (adjusted phenotype) by phenoconversion [33]. For the CYP2C19 PM phenotype, the rate increased from 2.7% (genetic phenotype) to 17% (adjusted phenotype) by taking phenoconversion due to DGIs into account.

In Drug–Drug–Gene-Interactions, the drug interaction becomes clinically relevant due to the genetic polymorphism. For example, if a CYP2D6 NM patient on metoprolol 50 mg per day receives a strong inhibitor for CYP2D6, the serum levels of metoprolol will probably increase fivefold. However, if the patient is a CYP2D6 PM, the levels of metoprolol will not change at all if a strong inhibitor for CYP2D6 is coprescribed. Citalopram is another prime example of a DDGI, as it is metabolized by both CYP2D6 and CYP2C19. In detail, if a patient is IM for CYP2D6 and NM for CYP2C19, they can receive a “normal” drug dose, because the main metabolic pathway is “open” (NM status). Now, if said patient receives a CYP2C19 inhibitor, such as omeprazole, the drug level will increase significantly due to the inhibition of the CYP2C19 and IM status on CYP2D6. Finally, if the patient is UM for CYP2D6 and receives the CYP2C19 inhibitor omeprazole, the patient would still have “normal” serum levels (Table 6 and Table 7).

Drug interactions (inhibition or induction) tend to be clinically relevant if the phenotype in the main metabolism pathway is of a poor or intermediate metabolizer status. Those patients are more sensitive to minor changes in usually less relevant pathways that occur if weak or moderate inhibitors are added to the drug therapy. For example, if a weak or moderate inhibitor of CYP3A4 is coadministered in a patient that receives a drug that is metaboized by CYP2D6 and CYP3A4 in combination and he has a CYP2D6-poor metabolizer status (main pathway). Such an interaction is more relevant than in a normal metabolizer patient, even though the same drugs are prescribed to both patients. There are a number of studies confirming that the relevance of the drug interaction depends on the genotype of the patient. Bahar showed in a PharmLine study how often, several pathways for the drug metabolism are affected by the drugs and genotypes [37]. In 55.1% of the patients, the study found a phenotype that required dose adaption; in 44.9%, he found a phenotype plus a drug interaction on a relevant pathway. He found that 9%, 47%, and 8.5% of participants were exposed to DDIs, DGIs, and DDGIs, respectively. Furthermore, there was an indication that the copresence of CYP3A4 IM/PM in individuals with CYP2C19 IM/PM exhibited an increased the risk of switching and/or dose reduction of (es)citalopram to a larger extent than the combination of CYP2C19 IM/PM and CYP3A4 NM (aOR: 4.38, 95% CI: 1.22–15.69 and aOR: 2.75, 95% CI: 1.03–7.29, respectively). DDGIs also seemed to increase the risk of drug switching and/or dose reduction (aOR: 2.33, 95% CI: 0.42–12.78).

Additionally, Storelli showed that CYP2D6 NMs carrying a nonfunctional allele are at particular risk of phenoconversion to a poor metabolizer status in the presence of CYP2D6 inhibitors [38]. Seventeen homozygous carriers of two fully functional alleles and 17 heterozygous carriers of one fully functional and one nonfunctional allele participated in Storelli´s analysis. Dextromethorphan 5 mg and tramadole 10 mg were applied at each of the three study sessions. CYP2D6 was inhibited by duloxetine 60 mg (session 2) and paroxetine 20 mg (session 3). A higher rate of phenoconversion to intermediate metabolizers with duloxetine (71% vs. 25%, *p* = 0.009) and to poor metabolizers with paroxetine (94% vs. 56%, *p* = 0.011) was observed in heterozygous compared with homozygous normal metabolizers. The magnitude of the DDI between dextromethorphan and paroxetine was higher in homozygous than in heterozygous subjects (14.6 vs. 8.5, *p* < 0.028). This strongly suggests that genetic normal metabolizers may not represent a homogenous population and that available genetic data should be considered when addressing DDIs in clinical practice. In concequence, the laboratories need to report genotypes and phenotypes.

Verbeurgt also analyzed the prevalence of DDIs, DGIs, and DDGIs and found 1053 potential major or substantial interactions in a cohort of 501 individuals. DDIs accounted for 66.1% of the total interactions. The remaining 33.9% of interactions were DGIs (14.7%) and DDGIs (19.2%). Interestingly, when compared with DDIs alone, DGIs and DDGIs increased the total number of potentially clinically significant interactions by 51.3%, showing how important PGx testing in clinical practice is [39]. In psychopharmacotherapy, Hefner et al. found that, in a cohort of 27,396 pyschiatric inpatients, 14.4% received a CYP inhibitor or inducer, opening the way for phenoconversion, as described in Table 6 and Table 7 [40]. The most frequently prescribed CYP inhibitors were melperone (*n*  =  2504, 28.1%) and duloxetine (*n*  =  1324, 14.9%). Overall, 51.0% of the cases taking melperone were combined with a victim drug (*n*  =  1288). Carbamazepine was the most frequently prescribed CYP inducer (*n*  =  733, 88.8%), and within those cases, a combination with victim drugs were detected for 58% (*n*  =  427). Finally, a relevant DDI was detected in 43.6% of the cases in which a CYP inhibitor or inducer was prescribed.

However, DDGIs may even be caused by a combination of substrates. Monte analyzed a cohort of patients receiving dextromethorpan, a CYP2D6 substrate. The coingestion of another CYP2D6-dependent drug was 9.49 (95% CI: 1.54, 186.41; *p* = 0.01) times more likely to have genotype–phenotype discordance based upon the 3-h DX/DM ratio. CYP2D6 substrate coingestions also caused genotype–phenotype discordance [41]. However, it is important to state that these findings need confirmation in other, larger (future) studies.

However, phenoconversion does not always result in a changed phenotype, e.g., if a poor metabolizer of CYP2C19 receives a CYP2C19 inducer, the phenotype remains PM, because more “loss of function proteins” will be synthesised, which does not increase the clearance of the drug. Moreover, if poor metabolizers receive a strong inhibitor for the same enzyme, there is no change in drug clearance either, since their phenotype does not change. Such an activity score–phenoconversion table does already exist for CYP2D6 and CYP2C19 (Table 6 and Table 7).

Predicting the phenotype from genotype in clinical practice for individualizing therapy becomes virtually impossible when individuals start taking commonly prescribed, and widely used, CYP2C19 inhibitors such as the PPI omeprazole. Klieber at al. recommended using a pantoprazole-13C breath test (Ptz-BT) to define the phenotype when coadministered with CYP inhibitors omeprazole and esomeprazole [42]. After 28 days of PPI therapy, a genotype–phenotype discordance was discovered in 27 out of 29 non-PM patients (93%). Interestingly, all intermediate metabolizers were converted to poor metabolizers, but the normal metabolizers were either converted to intermediate or poor metabolizers. The use of commonly and widely prescribed CYP2C19 inhibitors—omeprazole and esomprazole—led to a phenoconversion of CYP2C19 enzyme activity. Klomp et al. showed in their review that the effect of phenoconversion differs between individuals and between drugs. In detail, normal metabolizers are sometimes converted to IM or PM, which is probably due to DGGIs [43]. Van de Wouden et al., therefore, recommended a PGx passport for patients who were tested and to make the information available to all prescribers at all times [44].

## 9. Conclusions

Genetic variations markedly increase or ameliorate the severity of potential drug interactions and need to be considered when prescribing patients with polypharmacology. Preemptive PGx testing plus a drug–drug–gene interaction check to take phenoconversion into account could avoid adverse drug events. This could also improve the clinical recommendations (which drug to choose) and dose individualization for patients. It allows an assessment of the relevance of a pharmacokinetic drug interaction and, therefore, can help to minimize alert fatigue. It must be emphasized that only testing for certain CYP enzymes is a highly misleading strategy, because potential DGGIs could be missed. It is important to screen for a panel of SNPs, as proposed by DPWG and Bousman et al., when trying to predict a serum level of a drug and its efficacy and tolerability. Influx and eflux transporter polymorphisms, as well as phase I and II DME polymorphisms, influence the efficacy and tolerability of a drug. This increasing complexity of pharmacotherapy by polypharmacy, DDIs, DGI, DDGIs, and DGGIs makes a clinical pharmacist with skills in pharmacokinetics and PGx essential in a healthcare team in the future, if not even in the present. TDM, PGx testing, and drug interaction checks need to be combined to make valid interpretations of the pharmacokinetic profile. Clinical studies need to be aware of phenoconversion and take all factors into account when interpreting TDM results or PGx results. In the future, recognizing DGIs, DDGIs, and DGGIs may lead to a more comprehensive method of identifying individuals who are at risk for adverse drug reactions. This may lead to a better reputation and clear role of PGx testing in the future and allow a more profound interpretation of TDM results.

## Figures and Tables

**Figure 1 pharmaceuticals-14-00487-f001:**
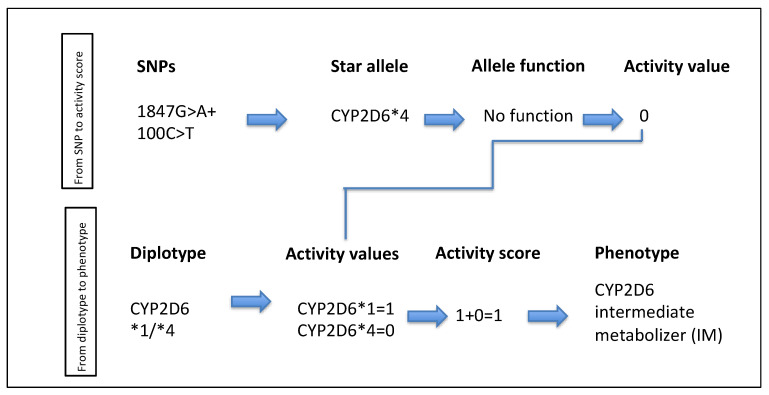
Explanation how to transfer a SNP to an activity score (first line) and how to calculate the phenotype from the diplotype (second line).

**Figure 2 pharmaceuticals-14-00487-f002:**
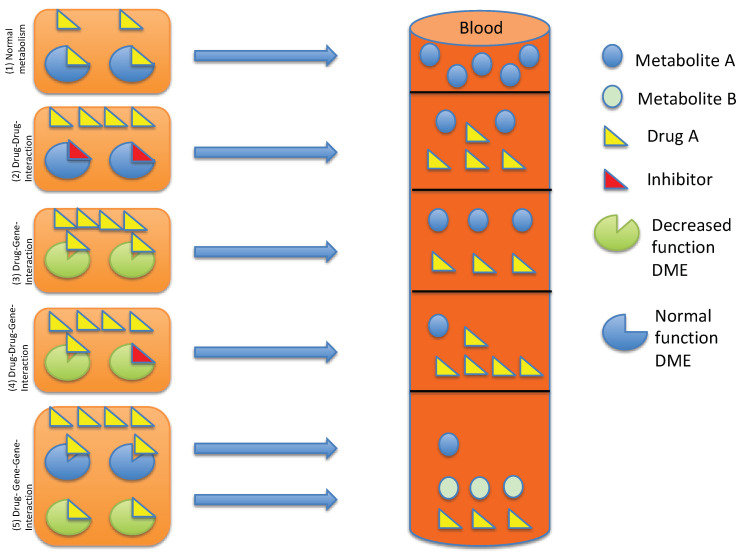
(1) Normal metabolism in normal metabolizers results in metabolism to metabolite A. (2) Drug–Drug-Interactions, e.g., a combination with an inhibitor of the drug-metabolizing enzyme, results in a decreased metabolism of the drug into metabolite A. The serum levels of drug A is increased compared to normal metabolism. (3) DGI: the phenotype of the drug-metabolizing enzyme determines the metabolism rate into an (active) metabolite, e.g., the phentoype IM leads to a decreased metabolism of the drug. A high drug concentration of the parent drug can be found by using TDM. (4) DDGI: an inhibitor or inducer of a drug-metabolizing enzyme changes the phenotype by phenoconversion. This changes the serum levels of the drug, e.g., increases the serum levels of the drug and decreases the levels of the metabolite A, e.g., if both the phenotype and perpetrator drug limit the drug metabolism, high serum levels of the parent drug can be found. (5) DGGI: the phenotype of two drug-metabolizing enzymes determines the formation of metabolite A and B, e.g., if the main pathway is “closed” due to a poor or intermediate metabolizer status, the phenotype of the drug-metabolizing enzyme of the second pathway determines the speed of the metabolism. Metabolite B is formed to a larger extent than metabolite A.

**Table 1 pharmaceuticals-14-00487-t001:** Definitions of DDIs, DDGI, DGGIs, and phenoconversion.

Term	Definition
Drug–Drug Interaction	When a drug in the individual’s regimen affects that individual’s ability to clear another drug.
Drug–Gene Interaction	When an individual’s genetic phenotype affects that patient’s ability to clear a drug.
Drug–Drug–Gene Interaction	When the individual’s genetic AND another drug in the individual’s regimen affects that individual’s ability to clear a drug.
Phenoconversion	Mismatch between the individual’s genotype- based prediction of drug metabolism and true capacity to metabolize drugs due to non-genetic factos (e.g., inflamation, pregnancy, liver failure, GFR, smoking, gender, and comedication).
Drug–Gene–Gene Interaction	Mismatch between the expected capacity to metabolize a drug that is caused by a second metabolizing (alternative pathway) enzyme’s genotype.
Victim Drug	Substrate of drug-metabolizing enzymes that are induced or inhibited in combination with a perpetrator drug (inhibitor or inducer). The serum levels of the vitim drug changes by this Drug–Drug-Interaction.
Perpetrator Drug	Inhibitor or inducer of drug-metabolizing enzymes that increases or decreases the serum levels of the victim drug. The serum level of the perpetrator drug does not change.

**Table 2 pharmaceuticals-14-00487-t002:** CYP enzymes, their phenotypes, substrates, and drugs that can cause phenoconversion by inhibition or induction. Underlined: CYP inducers. NM = normal metabolizers; IM = intermediate metabolizers; PM = poor metabolizers; RM = rapid metabolizers; UM = ultra-rapid metabolizers; NSAIDs = nonsteroidal antiinflammatory drugs. Examples from http://go.drugbank.com, accessed on 29 March 2021.

CYP	Known Phenotypes	Substrates	Phenoconversion
1A2	increased funtion normal function unknown function	duloxetine, olanzapin, clozapine, theophyllin, caffeine	fluvoxamine, ciprofloxacine, enoxacine, smoking
2A6	PM, IM, NM, UM	nicotine	
2B6	NM, IM, PM, RM, UM	bupropion, cyclophospamide, efavirenz, methadone	clopidogrel, ticlopidine, tenofovir, voriconazole, carbamazepine, efavirenz, rifampin
2C8	increased function normal function decreased function	glitazones, paclitaxel	gemfibrozil, clopidogrel, teriflunomide, trimethoprim, rifampin, St. John‘s wort
2C9	NM, IM, PM	losartan, NSAIDs, phenytoin, warfarin, glyburide	amiodarone, fluconazole, miconazole, rifampin
2C19	NM, IM, PM, RM, UM	clopidogrel, diazepam, proton pump inhibitors (PPI)	fluvoxamine, fluoxetine, fluconazole, omeprazole, ticlopidine, rifampin
2D6	NM, IM, PM, UM	antidepressants, betablockers, codeine, tramadol, tamoxifen, hydrocodone	bupropion, cimetidine, duloxetine, fluvoxamine, fluoxetine, paroxetine, quinidine, Note: there are no known inducers of CYP2D6.
3A4	normal function, decreased function, increased function	calcium channel blockers, macrolides, protease inhibitors, statins	azole antimycotics, boceprevir, cobicistat, danoprevir, grapefruit, ritonavir, telaprevir, verapamil, carbamazepine, phenobarbital, phenytoin, rifampin, St. John’s wort
3A5	NM, IM, PM Note: activity has major influence on CYP3A4 activity, if *1 is present	Tacrolimus, quetiapine	Ciprofloxacin, erythromycin, diltiazem, ketoconazole, verapamil

**Table 3 pharmaceuticals-14-00487-t003:** Examples of the activity scores of CYP2D6. ^a^
*CYP2D6*2* is currently considered to be a normal function allele by CPIC and DPWG; however, this function assignment has been challenged, and some laboratories report the *CYP2D6*2* function differently. The function of this allele will be reassessed as additional data become available. ^b^ N is categorical and indicates the number of copy variants (e.g., **1* × *2*, **1* × *3*, etc.).

Activity Score	Alleles (Examples)	Type of Allele and Genotype
>2.25	*1/*1 × N, *1/*2 × N ^b^*2 ^a^/*2 × N ^b^, *1 × 2/*9	Increased activity, Ultra rapid metabolizer
≤2.25 to ≥1.25	*1/*10, *1/*41, *1/*9, *1/*1, *1/*2, *2 × 2/*10	Wild-type, Normal metabolizer
>0 to <1.25	*4/*10, *4/*41, *10/*10, *10/*41, *41/*41, *1/*5	Reduced function, Intermediate metabolizer
0	**3/*4,*4/*4*,**5/*5*,**5/*6*	Non-functional, Poor metabolizer

**Table 4 pharmaceuticals-14-00487-t004:** Examples for phase 2 DME, phenotypes, substrates and inhibitors/inducers. N.a. = not applicable; NM = normal metabolizers; IM = intermediate metabolizers; PM = poor metabolizers; RM = rapid metabolizers; UM = ultra-rapid metabolizers. Examples are from the pharmgkb database: www.pharmgkb.org, accessed on 29 March 2021.

Enzyme	Known Phenotypes	Substrates	Phenoconversion
UGT1A1	NM, IM, PM	bilirubin, irinotecan, estradiol	Atazanavir, carbamazepine, phenytoin, phenobarbital, rifampicin, ritonavir, lamotrigin, efavirenz, tyrosine-kinase inhibitors
UGT1A4	Normal function, increased function, decreased function	valproic acid, lamotrigine, allopurinol, febuxostat, tamoxifen, clozapine, anastrozole	methylene blue, ertugliflozin, carbamazepine, phenytoin
UGT1A6	n.a.	allopurinol, febuxostat, methothrexat, valproic acid	troglitazone, fosphenytoin, phenytoin, carbamazepine
UGT1A9	n.a.	allopurinol, febuxostat, methothrexat, valproic acid	vandetanib
UGT2B7	n.a.	zodovudine, oxycodone, efavirenz, methadone, lamotrigine, morphine, codeine, fentanyl.	flunitrazepam, ketoconazole, umifenovir, phenobarbital, mefenamic acid
UGT2B15	normal function decreased funtion	oxazepam, lorazepam	
N-acetyltransferase (*NAT2*)	fast slow	isoniazid, hydralazine, dapsone, caffein, procainamide	
Thiopurine Methyl Transferase *(TPMT)*	NM, IM, possibly intermediate, PM	thiopurines	allopurinol
Nudix hydrolase 15 (*NUDT 15*)	NM, IM, possibly intermediate, PM	thiopurines	

**Table 5 pharmaceuticals-14-00487-t005:** Examples for influx and efflux transporters, their genotypes, and examples of the substrates and interacting drugs causing phenoconversion. Underlined drugs: inducers for the transporter. Examples were retrieved from the pharmgkb database: www.pharmgkb.org, accessed on 29 March 2021.

Gene/Transporter	Known Phenotypes	Substrates	Phenoconversion
OATP1B1/*SLCO1B1 gene*	normal function, decreased function, poor function	atorvastatin, repaglinide, enalapril, methotrexate, rosuvastatin, simvastatin, eryhtromycin, nateglinide, pitavastatin, pravastatin, lopinavir	astemizole, diazepam, nifedipine
BCRP/*ABCG2 gene*	Normal function, decreased function	allopurinol, asuvastatin, leflunomide, sunitinib, topotecan, pitavastatin, rosuvastatin, sulfasalazine	curcumine, elacridar, cyclosporine A
P-glycoprotein/*ABCB1/MDR1 gene*	normal function, increased function	colchicine, fexofenadine, simvastatin, rifampin, cyclosporine, ondansetron, risperidone, digoxin, fentanyl, methadone, oxycodone, tramadole, phenytoin	amiodarone, carvedilol, clarithromycin, quinidine, verapamil, ritonavir, telaprevir, carbamazepine, St. John’s wort, primidone, rifampin, phenytoin

**Table 6 pharmaceuticals-14-00487-t006:** Phenoconversion in CYP2D6, and the calculation of the activity scores and the resulting phenotype.

Activity Score CYP2D6	Genetic Phenotype	Weak Inhibitor and Moderate Inhibitor	Strong Inhibitor
0	PM	Activity score × 0.5 = PM	Activity score × 0 = PM
> 0 < 1.25	IM	Activity score × 0.5 = IM	Activity score × 0 = PM
> 1.25 < 2.25	NM	Activity score × 0.5 = IM	Activity score × 0 = PM
>2.25	UM	Activity score × 0.5 = NM	Activity score × 0 = PM

**Table 7 pharmaceuticals-14-00487-t007:** Phenoconversion in CYP2C19, and the calculation of the activity scores and the resulting phenotype.

Genetic Phenotype CYP2C19	Comedication of a Moderate or Strong Inhibitor; Predicted Phenotype
NM, IM	PM
RM, UM	IM
PM	PM
	Comedication of a moderate or strong inducer; Predicted phenotype
NM, RM	UM
IM	NM
PM	PM
UM	UM

## Data Availability

Not applicable.

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
