# Peer review of "The Influence of Pharmacogenetics on the Clinical Relevance of Pharmacokinetic Drug–Drug Interactions: Drug–Gene, Drug–Gene–Gene and Drug–Drug–Gene Interactions"

_pharmaceuticals, 2021, doi:10.3390/ph14050487_

Round 1
Reviewer 1 Report
Thank you to the author for submitting the paper for revision. While the paper does touch on important facets of expanding investigations into the clinical relevance for drug-drug-gene and drug-gene-gene interactions, there are some glaring points which need to be improved.
- The number of spelling mistakes and grammatic errors throughout the paper is high. Some mistakes could have been avoided with a spell check but others include the misspelling of important names (e.g. DPWG was spelt DWPG). Please go over the paper carefully and double-check these mistakes. Also the citation format seems different per article in the References section
- The structure of the paper is quite jumbled. While the sections provide a semblance of structure for the paper, some sections can be merged with other sections (as sub-sections) to provide clearer explanation of DDIs, DGIs, etc. Using the example of section 6, there is only a table therefore either supportive text explaining what the table signifies or how it relates to DGIs would ensure better legibility of the paper.
- Of the sections which are of interest (e.g. DGGI/DDGIs), Section 8 is lacking in information or rather provides only a glimpse of what relevant clinical opportunities there are for incorporating monitoring of such interactions
The paper below is quite an interesting read and touches on what you are reviewing in this paper. Hopefully it helps with the refining of this section.
Bechtold B, Clarke J. Multi-factorial pharmacokinetic interactions: unraveling complexities in precision drug therapy. Expert Opin Drug Metab Toxicol. 2021 Jan 20:1-16. doi: 10.1080/17425255.2021.1867105. Epub ahead of print. PMID: 33339463.
Author Response
Reviewer 1:
Thank you to the author for submitting the paper for revision. While the paper does touch on important facets of expanding investigations into the clinical relevance for drug-drug-gene and drug-gene-gene interactions, there are some glaring points which need to be improved.
- The number of spelling mistakes and grammatic errors throughout the paper is high. Some mistakes could have been avoided with a spell check but others include the misspelling of important names (e.g. DPWG was spelt DWPG) Corrected. Please go over the paper carefully and double-check these mistakes. Also the citation format seems different per article in the References section Done, review by native speaker (Prof. Slattery, Frankfurt) done.
- The structure of the paper is quite jumbled. While the sections provide a semblance of structure for the paper, some sections can be merged with other sections (as sub-sections) to provide clearer explanation of DDIs, DGIs, etc. Using the example of section 6, there is only a table therefore either supportive text explaining what the table signifies or how it relates to DGIs would ensure better legibility of the paper. Sections „Genetic polymorhisms“, „phase 1“, „2“ and „3 metabolising enzymes“ are integrated in one chapter of „Drug-Gene Interactions“.
- Of the sections which are of interest (e.g. DGGI/DDGIs), Section 8 is lacking in information or rather provides only a glimpse of what relevant clinical opportunities there are for incorporating monitoring of such. interactions Information is added and clinical relevance stated in the conclusion section.
The paper below is quite an interesting read and touches on what you are reviewing in this paper. Hopefully it helps with the refining of this section.
Bechtold B, Clarke J. Multi-factorial pharmacokinetic interactions: unraveling complexities in precision drug therapy. Expert Opin Drug Metab Toxicol. 2021 Jan 20:1-16. doi: 10.1080/17425255.2021.1867105. Epub ahead of print. PMID: 33339463.
Reviewer 2 Report
In their manuscript „The Influence of Pharmacogenetics on the Clinical Relevance of Drug-Drug-Interactions: Drug-Gene-, Drug-Gene-Gene-, and Drug-Drug-Gene-Interactions” the authors summarized the current state of knowledge on the relevance of the genetic background of metabolizing enzymes and transporters on drug therapy. Despite the fact that important aspects are missing, several improvements in the overall structure of the manuscript should be performed.
- Overall, this article is designed as a review providing an overview about a certain topic to readers not deeply involved in the respective field. Therefore, I strongly suggest explaining the important subjects at the beginning of the article and not in table 7. Looking in Pubmed entering e.g. the title words drug-gene-gene interaction or drug-drug-gene interaction there are just one and 32 hits, respectively demonstrating that these denotations are relatively new and should be explained in detail (at best with good examples). On the other hand, detailed explanations regarding the definition of genetic polymorphisms (Chapter 3) is not necessary and can be shortened.
- What was the reason for mentioning SLC transporters but not including them into table 4? At least for the SLC21 or SLC22 family members there are clinical data regarding the role of genetic variants on drug therapy. That has to be included.
- A table with important genetic variants of metabolizing enzymes and transporters and their impact on drug pharmacokinetics and –dynamics should be included.
- Paragraph one from the Abstract and the Introduction are identical (including the same typo). This is not usual and should be changed.
- Please include a detailed legend for figure 2.
Minor:
Abstract (line 12) and Introduction (line 25): with instead of whith
Abstract (line 16): please change “The clinical relevance of DDIs depends on….” Into “The clinical relevance of DDIs also depends on….”
Keywords: pharmacogenetics instead of pharmakogenetics
Line 75: each other
Line 96: serum concentrations of the victim drugs
Page 5, line 164: to the best of the reviewer´s knowledge for at least CYP2D6 there are just 4 (not 5) different genetic phenotypes: poor metabolizer (PM), intermediated metabolizer (IM), extensive metabolizer (EM) and ultrarapid metabolizer (UM) (see e.g. Zanger et al., (2004) Naunyn-Schmiedeberg´s Archives of Pharmacology 369). This has to be clarified.
Line 185: inhibitors
Page 8, Table 4: Gene/Transporter in column one is missing (maybe OATP1B1?)
Line 251: There are two major transporter superfamilies: ABC transporter and SLC transporter consisting of ABC transporter and SLC transporter families (e.g. ABCC family or SLC21 family). – please correct.
Line 256: A prominent example is that statin induced myopathy in carriers of the OATP1B1*5 polymorphism – please correct.
Line 280 (and whole manuscript): SLCO1B1 and ABCG2 (in italics) genotypes
Line 429: efflux instead of eflux
Line 437: and instead of ans
Author Response
Reviewer 2:
Thank you for your remarks to our manuscript. They were very helpful.
The purpose of this review is to discuss the importance of the genetic profile of the patients when analysing the potential side effects that might appear when taking multiple drugs. This specific subject is of great importance and writing a review to cover some aspects in details is welcome. However, the current version of the article suffers from various issues which need to be solved before it can be considered for publication.
General comments about the pictures: The two pictures provided are of poor quality and suffer from editorial flaws. More precisely, there are unwanted symbols “?” appearing near the terms of the legend of Fig. 2 and near almost all terms written within Fig 1. The Fig. 1 should be redone because the graphics is poorly designed, the text on the two boxes on the left sides cannot be read. Most of the space taken by the picture is empty white space. Maybe the authors should add graphical representation illustrating the concepts written in the different parts of the pictures. For both pictures, the authors should add a more detailed legend of what the pictures represent (normally the legend should allow the reader to grasp the concepts summarized within the pictures without referring to the main text.) and how this articulates with what is described in the text and support it. A legend is added, the poor design remark is unclear. Maybe this is more an issue of reformatting by using different programs or version to open the file? In my proofs, everything is readable and well structured. Also the feedback from the other three reviewers is lacking a comment about he figure, maybe only a formatting problem of the editorial manager website?
Comments about the introduction. The first paragraph of section 1 was copy pasted from the abstract (or it is the introduction that was taken to make the abstract). Normally the abstract should give a short overview of the background, motivation and key points covered in the review but it should not be taken from the introduction or any other parts of the main text. Rewritten.Here the abstract does not even mention the CYP-inhibitors that are extensively covered and discussed in most of the rest of the paper. Done In the introduction, the authors introduce many concepts of importance for what is discussed in the article, but the articulation and structure of the introduction should be improved so the reader understands for instance that CYP-inhibitors play a major role in what is about to be discussed. The last paragraph of the introduction could contain a short summary of each key section of the article so the reader understands the overall structure of the review. Done
The abstract and introduction were partly rewritten. Done
General note about abbreviations. There are some abbreviations not defined (see ST line 41) and others that are defined multiple times (for instance, DDI defined a first time line 29 and redefined line 71). Abbreviations should be defined the first time they appear within the mean text. The abstract should not contain abbreviations. Done
There are many typos and many grammatical mistakes. I would recommend carefully checking the article for grammar and spelling.Reviewed by native speaker. (Prof. Slattery, Frankfurt)
Line 16: “DDIs depends on the genetic profile of the patient” would be more appropriate Done
Line 17: “This review discusses the current evidence” Done
Line 24-29: it is the abstract word by word… Abstract rewritten.
Line 42: 98% of the patients Done
Line 49-50 the sentence is not clear, please rephrase. Rephrased.
Line 79 cannot Done
Line 89: its Done
Line 133: one can Done
Line 135: It is made of two or more allels Done
Line 151-152: those abbreviations have already been defined Done
Line 293: therefore Done
Line 297 each other Done
Line 395: does not Done
Line 437 “ans” should be and Done
Reviewer 3 Report
The purpose of this review is to discuss the importance of the genetic profile of the patients when analysing the potential side effects that might appear when taking multiple drugs. This specific subject is of great importance and writing a review to cover some aspects in details is welcome. However, the current version of the article suffers from various issues which need to be solved before it can be considered for publication.
General comments about the pictures: The two pictures provided are of poor quality and suffer from editorial flaws. More precisely, there are unwanted symbols “?” appearing near the terms of the legend of Fig. 2 and near almost all terms written within Fig 1. The Fig. 1 should be redone because the graphics is poorly designed, the text on the two boxes on the left sides cannot be read. Most of the space taken by the picture is empty white space. Maybe the authors should add graphical representation illustrating the concepts written in the different parts of the pictures. For both pictures, the authors should add a more detailed legend of what the pictures represent (normally the legend should allow the reader to grasp the concepts summarized within the pictures without referring to the main text.) and how this articulates with what is described in the text and support it.
Comments about the introduction. The first paragraph of section 1 was copy pasted from the abstract (or it is the introduction that was taken to make the abstract). Normally the abstract should give a short overview of the background, motivation and key points covered in the review but it should not be taken from the introduction or any other parts of the main text. Here the abstract does not even mention the CYP-inhibitors that are extensively covered and discussed in most of the rest of the paper. In the introduction, the authors introduce many concepts of importance for what is discussed in the article, but the articulation and structure of the introduction should be improved so the reader understands for instance that CYP-inhibitors play a major role in what is about to be discussed. The last paragraph of the introduction could contain a short summary of each key section of the article so the reader understands the overall structure of the review.
General note about abbreviations. There are some abbreviations not defined (see ST line 41) and others that are defined multiple times (for instance, DDI defined a first time line 29 and redefined line 71). Abbreviations should be defined the first time they appear within the mean text. The abstract should not contain abbreviations.
There are many typos and many grammatical mistakes. I would recommend carefully checking the article for grammar and spelling.
Line 16: “DDIs depends on the genetic profile of the patient” would be more appropriate
Line 17: “This review discusses the current evidence”
Line 24-29: it is the abstract word by word…
Line 42: 98% of the patients
Line 49-50 the sentence is not clear, please rephrase.
Line 79 cannot
Line 89: its
Line 133: one can
Line 135: It is made of two or more allels
Line 151-152: those abbreviations have already been defined
Line 293: therefore
Line 297 each other
Line 395: does not
Line 437 “ans” should be and

Author Response
Reviewer 3: thank you for your positive feedback on our article.
Dear Editor,
According to the guidelines for reviewers, I would like to report on my reviewing progress.
I recommend the manuscript “The Influence of Pharmacogenetics on the Clinical Relevance of Drug-Drug-Interactions: Drug-Gene-, Drug-Gene-Gene- and Drug-Drug-Gene-Interactions” to be accepted for publication.
The manuscript is very comprehensive, up to date and includes the relevant and new information both for clinicians and basic researchers.
I can strongly recommend this manuscript to be published.
Regarding the originality of this manuscript, the article is novel and interesting.
The research question of the article is valid and an important one.
This article is in the top 5% of papers in this field.
The abstract is an accurate summary of reviewed data.
The language of the manuscript is clear and understandable.
Reviewer 4 Report
Dear Editor,
According to the guidelines for reviewers, I would like to report on my reviewing progress.
I recommend the manuscript “The Influence of Pharmacogenetics on the Clinical Relevance of Drug-Drug-Interactions: Drug-Gene-, Drug-Gene-Gene- and Drug-Drug-Gene-Interactions” to be accepted for publication.
The manuscript is very comprehensive, up to date and includes the relevant and new information both for clinicians and basic researchers.
I can strongly recommend this manuscript to be published.
Regarding the originality of this manuscript, the article is novel and interesting.
The research question of the article is valid and an important one.
This article is in the top 5% of papers in this field.
The abstract is an accurate summary of reviewed data.
The language of the manuscript is clear and understandable.
Author Response
Reviewer 4
In their manuscript „The Influence of Pharmacogenetics on the Clinical Relevance of Drug-Drug-Interactions: Drug-Gene-, Drug-Gene-Gene-, and Drug-Drug-Gene-Interactions” the authors summarized the current state of knowledge on the relevance of the genetic background of metabolizing enzymes and transporters on drug therapy. Despite the fact that important aspects are missing, several improvements in the overall structure of the manuscript should be performed.
- Overall, this article is designed as a review providing an overview about a certain topic to readers not deeply involved in the respective field. Therefore, I strongly suggest explaining the important subjects at the beginning of the article and not in table 7. Looking in Pubmed entering e.g. the title words drug-gene-gene interaction or drug-drug-gene interaction there are just one and 32 hits, respectively demonstrating that these denotations are relatively new and should be explained in detail (at best with good examples). On the other hand, detailed explanations regarding the definition of genetic polymorphisms (Chapter 3) is not necessary and can be shortened. Table 7 is now table one in the introduction. Definition of genetic polymorphisms is shortened.
- What was the reason for mentioning SLC transporters but not including them into table 4? At least for the SLC21 or SLC22 family members there are clinical data regarding the role of genetic variants on drug therapy. That has to be included. It was included in table 4 (first line) in our final article already.
- A table with important genetic variants of metabolizing enzymes and transporters and their impact on drug pharmacokinetics and –dynamics should be included. A reveral to CPICs website is included in the section DGI. There are 440 genes (and many more genetic variants) and their impact listed which is to long to integrate in this review article. The article only focuses on pahrmacokinetic aspects and many examples are given in the text. The CPIC website is cited to enable the reader to access the list easily.
- Paragraph one from the Abstract and the Introduction are identical (including the same typo). This is not usual and should be changed. Abstract rewritten.
- Please include a detailed legend for figure 2. A Legend was written for both figure 1 and 2.
Minor:
Abstract (line 12) and Introduction (line 25): with instead of whith DONE
Abstract (line 16): please change “The clinical relevance of DDIs depends on….” Into “The clinical relevance of DDIs also depends on….” DONE
Keywords: pharmacogenetics instead of pharmakogenetics DONE
Line 75: each other DONE
Line 96: serum concentrations of the victim drugs DONE
Page 5, line 164: to the best of the reviewer´s knowledge for at least CYP2D6 there are just 4 (not 5) different genetic phenotypes: poor metabolizer (PM), intermediated metabolizer (IM), extensive metabolizer (EM) and ultrarapid metabolizer (UM) (see e.g. Zanger et al., (2004) Naunyn-Schmiedeberg´s Archives of Pharmacology 369). This has to be clarified. CPIC defined a rapid metabolizer phenotype for CYP2C19 (*1/*17). Please refer to the CPIC guideline on PPI use Lima et al. 2020.
Line 185: inhibitors ? Done
Page 8, Table 4: Gene/Transporter in column one is missing (maybe OATP1B1?) ? in the proofs I can see OATP1B1 written in column one?
Line 251: There are two major transporter superfamilies: ABC transporter and SLC transporter consisting of ABC transporter and SLC transporter families (e.g. ABCC family or SLC21 family). – please correct. Done
Line 256: A prominent example is that statin induced myopathy in carriers of the OATP1B1*5 polymorphism – please correct. Done
Line 280 (and whole manuscript): SLCO1B1 and ABCG2 (in italics) genotypes Done
Line 429: efflux instead of efluxm Done
Line 437: and instead of ans Done
Round 2
Reviewer 2 Report
The authors have addressed some thoughts and suggestions. Nevertheless, there are several unclear points and inconsistencies that weaken this manuscript.
In the available version both figures are of bad quality (e.g. question marks in Fig. 1 without meaning)
Fig. 2 is confusing and hard to understand. Labeling mentioned in the legend cannot be found in the figure (e.g. (1) DGI – where is (1)?)
Table 2: what does “induction incertain genotypes” mean? Please include legend explaining the different abbreviations
Table 3: empty part with just “Pharmacokinetic” ?
In table 4 – OATP1B1 the authors mentioned as phenotypes: decreased function and poor function. The difference between “decreased” and “poor” function should be explained. Furthermore, references should be included.
Minor: The revised version was written very superficially and contains a large amount of typos and inconsistencies. Here are some of them:
Title and whole manuscript: Pharmacokinetic instead of pharmakokinetic
Line 87: with instead of whith
Table 3: Phase instead of Pahse and inhibitors instead of inhibotrs
Table 4: Heading: Influx transporters, their genotypes…. But in the table the only “influx” transporter mentioned is OATP1B1 – BCRP and MDR1 are efflux pumps
Table 4: column Gene/Transporter: SLCO1B1 gene, ABCG2 gene and P-glycoprotein without coding gene – this has to be consistent
Line 240: What does (ABC tenofovir tubular toxicity… ) mean?
Line 275: TPMT instead of TMPT
Author Response
Reviewer 2 Thank you for your feedback!
The authors have addressed some thoughts and suggestions. Nevertheless, there are several unclear points and inconsistencies that weaken this manuscript.
In the available version both figures are of bad quality (e.g. question marks in Fig. 1 without meaning) not in my version. Dear Editor please check for malfunction of the submission platform. Reviwer 3 also had problems, but could solve the issue by opening the file in a different browser.
Fig. 2 is confusing and hard to understand. Labeling mentioned in the legend cannot be found in the figure (e.g. (1) DGI – where is (1)?) Labeling was added in the figure 2, the legend was rewritten.
Table 2: what does “induction incertain genotypes” mean? Please include legend explaining the different abbreviations „induction in certain genotypes“ was deleted. Abbreviations (UM, PM etc) are explained in the legend now.
Table 3: empty part with just “Pharmacokinetic” ? not in my version? (see below)
Activity score |
Alleles (examples) |
Type of allele and genotype |
>2.25 |
*1/*1xN, *1/*2xNb*2 a/*2xN b, *1x2/*9
|
Increased activity, Ultra rapid metabolizer
|
≤2.25 to ≥1.25 |
*1/*10, *1/*41, *1/*9, *1/*1, *1/*2, *2x2/*10 |
Wild-type, Normal metabolizer |
> 0 to < 1.25 |
*4/*10, *4/*41, *10/*10, *10/*41, *41/*41, *1/*5 |
Reduced function, Intermediate metabolizer |
0 |
*3/*4,*4/*4,*5/*5,*5/*6
|
Non-functional, Poor metabolizer |
In table 4 – OATP1B1 the authors mentioned as phenotypes: decreased function and poor function. The difference between “decreased” and “poor” function should be explained. Furthermore, references should be included. The whole paragraph with the explanation of floss of function, decreased function enzymes etc is in the „Drug-Gene Interaction“ paragraph.
Minor: The revised version was written very superficially and contains a large amount of typos and inconsistencies. Here are some of them:
Title and whole manuscript: Pharmacokinetic instead of pharmakokinetic was changed
Line 87: with instead of whith was changed
Table 3: Phase instead of Pahse and inhibitors instead of inhibotrs was changed
Table 4: Heading: Influx transporters, their genotypes…. But in the table the only “influx” transporter mentioned is OATP1B1 – BCRP and MDR1 are efflux pumps was changed to influx and efflux transporters
Table 4: column Gene/Transporter: SLCO1B1 gene, ABCG2 gene and P-glycoprotein without coding gene – this has to be consistent, it was changed accordingly to ABCB1 gene.
Line 240: What does (ABC tenofovir tubular toxicity… ) mean? Was changed to: „tenofovir induced tubular toxicity in patients with an ABCC2 polymorphism“
Line 275: TPMT instead of TMPT was changed accordingly
Reviewer 3 Report
I went through the revised manuscript. Although the authors have made most of the required corrections, and have improved the overall content of the manuscript, I would recommend that the paper should undergo a careful proofreading for the english (just in the title with "pharmakokinetic" in place of pharmacokinetic). For the pictures, there are still discrepancies that appear when opening the pdf using adobe reader, but I am not if this is due to a software malfunction or to the picture themselves. Because those unwanted symbols do not appear when opening the document directly through firefox for instance.
Author Response
Reviewer 3 thank you for your feedback! I informed the editor about the problems with certain browsers.
I went through the revised manuscript. Although the authors have made most of the required corrections, and have improved the overall content of the manuscript, I would recommend that the paper should undergo a careful proofreading for the english (just in the title with "pharmakokinetic" in place of pharmacokinetic). For the pictures, there are still discrepancies that appear when opening the pdf using adobe reader, but I am not if this is due to a software malfunction or to the picture themselves. Because those unwanted symbols do not appear when opening the document directly through firefox for instance.
The manuscript was reviewed by Prof. David Slattery (native speaker).